# Experiences with Glofitamab Administration following CAR T Therapy in Patients with Relapsed Mantle Cell Lymphoma

**DOI:** 10.3390/cells11172747

**Published:** 2022-09-02

**Authors:** Alexander D. Heini, Ulrike Bacher, Naomi Porret, Gertrud Wiedemann, Myriam Legros, Denise Stalder Zeerleder, Katja Seipel, Urban Novak, Michael Daskalakis, Thomas Pabst

**Affiliations:** 1Department of Medical Oncology, Inselspital, Bern University Hospital, 3010 Bern, Switzerland; 2Department of Hematology and Central Hematology Laboratory, Inselspital, Bern University Hospital, 3010 Bern, Switzerland

**Keywords:** mantle cell lymphoma (MCL), CAR T, expansion, relapse, glofitamab

## Abstract

Mantle cell lymphoma (MCL) is a rare type of B-cell Non-Hodgkin lymphoma (NHL) affecting predominantly male patients. While complete remissions following first-line treatment are frequent, most patients ultimately relapse, with a usually aggressive further disease course. The use of cytarabine-comprising induction chemotherapy and autologous stem cell transplantation, Rituximab maintenance, Bruton’s tyrosine kinase (BTK) inhibitors and CAR T therapy has substantially improved survival. Still, options for patients relapsing after CAR T therapy are limited and recommendations for the treatment of these patients are lacking. We report two cases of patients with mantle cell lymphoma who relapsed after CAR T therapy and were treated with the bispecific CD20/CD3 T cell engaging antibody glofitamab. Both patients showed marked increases of circulating CAR T cells and objective responses after glofitamab administration. Therapy was tolerated without relevant side effects in both patients. One patient completed all 12 planned cycles of glofitamab therapy and was alive and without clinical progression at the last follow-up. The second patient declined further treatment after the first cycle and succumbed to disease progression. We review the literature and investigate possible mechanisms involved in the observed responses after administration of glofitamab, such as proliferation of CAR T cells, anti-tumor effects of the bispecific antibody and the role of other possibly contributing factors. Therapy with bispecific antibodies might offer an effective and well-tolerated option for patients with mantle cell lymphoma relapsing after CAR T therapy.

## 1. Introduction

Mantle cell lymphoma (MCL) is a rare B-cell lymphoma affecting predominantly male patients with a median age of 68 years at diagnosis [1]. While patients frequently achieve complete remissions following first-line treatment, most patients will ultimately relapse, with a usually aggressive further course of the disease. The use of cytarabine-comprising induction chemotherapy and subsequent autologous stem cell transplantation (ASCT) [2], of Rituximab maintenance [3], and of Bruton’s tyrosine kinase (BTK) inhibitors [4] has substantially improved survival rates. However, the outcome of patients relapsing after treatment with BTK inhibitors is dismal, with poor response rates and short duration of response to salvage therapy [5]. CD19 directed chimeric antigen T cell (CAR T) therapy has been shown to achieve unprecedented responses in this group of patients with an overall response rate of 93% in the pivotal trial [6,7]. Whereas a majority of patients achieve durable remissions, options for patients relapsing after CAR T therapy are limited and recommendations for the treatment of these patients are currently lacking.

Bispecific T cell engaging antibodies are antibody constructs with two or more antigen binding domains, one recognizing a tumor antigen, such as the B cell maturation antigen (BCMA), CD19 or CD20, and the other recognizing CD3 on T cells, which is essential for T cell receptor-mediated T cell activation [8]. Several antibody constructs targeting CD20 and the CD3ε subunit, such as mosunetuzumab [9], odronextamab [10] or glofitamab [11,12], have shown promising activity in relapsed or refractory (R/R) lymphomas.

Glofitamab has been studied in a phase I dose-escalation study in 171 patients with R/R B cell lymphomas, where it showed an overall response rate (ORR) of 65.7% in the phase II dose group [11]. In a cohort of 29 patients with R/R MCL, glofitamab showed an ORR of 81%. Of the 29 patients, 20 (69%) had had previous therapy with Bruton’s tyrosine kinase (BTK) inhibitors [12]. Odronextamab was studied in a phase I dose-escalation study with 145 patients with CD20 positive R/R B cell malignancies who were previously treated with CD20-directed antibody therapies. Among all 145 heavily pretreated patients, ORR was 51% with a substantially higher response rate in patients with follicular lymphoma (91%). In another study, of 30 patients with diffuse large B cell lymphoma (DLBCL), who had been previously treated with CAR T therapy, the ORR was 33% and the complete response (CR) rate was 27% [10]. In another phase I dose-escalation study, mosunetuzumab was studied in 236 patients with R/R B cell lymphomas. ORR was 34.9% in patients with aggressive lymphomas and 66.2% in patients with indolent lymphomas [13]. Similar response rates were demonstrated in a cohort of 90 patients with R/R follicular lymphoma after two or more previous lines of therapy. ORR was 78.9% and CR rate was 57.8%. In this study, 3 out of 90 patients (3.3%) were previously treated with CAR T therapy [9]. Treatment with CD20/CD3 bispecific antibodies is generally well tolerated. The most common side effects include neutropenia (mosunetuzumab 28.4%) and cytokine release syndrome (CRS) (mosunetuzumab 44%, glofitamab 58.6%) with higher grade CRS being rare (<10%).

Data on the use of CD20/CD3 bispecific antibodies in patients failing after CAR T therapy are scarce. In a recent study including 9 patients with DLBCL, who relapsed after CAR T therapy and were treated with glofitamab, an ORR of 67% was reported. In this study, CRS rate was substantially lower (22%) than previously documented [14]. To the best of our knowledge, no data have been reported on the use of glofitamab in patients with MCL who relapsed after CAR T therapy. In this study, we report on two patients with MCL, who relapsed after CAR T therapy, and showed clinically relevant responses after treatment with glofitamab.

## 2. Patients

Both patients gave written informed consent for their data to be reported. Patient #1 was first treated at our center in March 2017 and is still seen at regular intervals in our outpatient clinic. Patient #2 was treated between February 2021 and January 2022 and was being followed up in a private practice before and after. Case descriptions were written after electronic chart review. A last follow up was conducted on 31 March 2022.

## 3. Methods

The expansion and persistence of CAR T cells in the peripheral blood after Tecartus^®^ CAR T infusion was monitored by specific droplet digital PCR (ddPCR). The ddPCR analysis was performed as previously described [15] with a CAR T specific assay targeting the intracellular junction sequence between the effector CD3ζ and the costimulatory CD28 domain, as initially reported for qPCR-based copy number quantification [16] and RPP30 (ribonuclease P protein subunit 30) to normalize for quantification of CAR T copy numbers per µg DNA.

For flow cytometry analyses, samples were first analyzed using the lymphocyte screening tube (previously Cytognos, Salamanca Spain now Becton Dickinson Biosciences, San Jose, CA, USA) [17]. This panel includes B-cell (CD19, CD20), T-cell (CD3, CD4, CD8, CD5), NK cell (CD56), and clonality markers (light chains kappa/lambda). For further characterization, the additional lymphoma panel was used, containing the markers CD79b, CD23, CD22, CD10, CD200, FMC7 and LAIR1 (Becton Dickinson Biosciences, BD, San Jose, CA, USA). We analyzed samples on a FACS Canto II followed by the Lyrics platform, and we used FACSDiva version 6.1.3 and FACSuite version 1.4 (both by Becton Dickinson Biosciences, BD, San Jose, CA, USA) as previously described [18].

For flow cytometric detection and quantification of CAR T cells, we used the CD19 CAR Detection Reagent (Biotin), that detects CD19 CAR T cells via recognition of the CD19 CAR molecule, and that can be identified with PE labelled anti-Biotin-Antibodies (Miltenyi Biotec, Bergisch-Gladbach, Germany). The tube additionally contained an antibody panel for CD45, 7-AAD, CD3, CD4, CD8. A minimum threshold of 10 CAR T cell specific events was required to call a sample positive for CAR T cells. Results were documented as percentages of live cells. Experiments were done on the Lyrics platform and the FACSuite software was used as previously described [19].

## 4. Patient Case Descriptions

We report on two patients with MCL relapsing after CAR T treatment [20]. Patient characteristics, disease courses and response to previous therapies are summarized in Table 1, whereas response to glofitamab treatment is summarized in Table 2 and Figure 1.

### 4.1. Patient #1

The first patient was a 60-year-old female diagnosed with blastoid variant of mantle cell lymphoma without *TP53* mutation. Initial disease stage was IIB with mediastinal, infraclavicular and cervical lymphadenopathy. First line treatment consisted of six alternating cycles of rituximab, cyclophosphamide, doxorubicin, vincristine and prednisone (R-CHOP) and rituximab, dexamethasone, cytarabine and cisplatin (R-DHAP) chemotherapy followed by high-dose chemotherapy (HDCT) with bendamustine, etoposide, cytarabine and melphalan (BeEAM) and autologous stem cell transplantation (ASCT). The patient achieved a complete remission (CR) by PET/CT three months after HDCT/ASCT and maintenance therapy with rituximab was initiated. 21 months after HDCT/ASCT. The patient relapsed in her left breast, with a biopsy confirming CD20+, weak CD19+, weak CD22+, CD10+, weak CD5+ blastoid variant MCL. She underwent six cycles of bortezomib and ibrutinib treatment within the SAKK36/13 trial [21] and, again, achieved CR, and maintenance therapy with ibrutinib was started.

Fourteen months after initiation of ibrutinib maintenance, MRI revealed three new lesions in the left breast, and biopsy confirmed relapsing blastoid MCL. Peripheral lymphocytosis with flow cytometric evidence of lymphoma cells with expression of CD19, CD20, CD10, and kappa light chain restriction prompted a bone marrow examination, which identified 80% infiltration by lymphoma cells with identical immunophenotype. The patient was accepted for CAR T therapy within the early access program for brexucabtagene autoleucel (Tecartus^®^) and bridging therapy with rituximab and bendamustine was initiated with partial remission of lymphadenopathy and reduction of lymphocytosis. CAR T treatment was tolerated without evidence of emerging cytokine release syndrome (CRS) or immune cell associated neurotoxicity syndrome (ICANS) and, consequently, no anti-cytokine therapy (tocilizumab and/or steroids) was needed. A bone marrow sample taken 14 days after CAR T reinfusion showed no signs of MCL infiltration, complete B cell aplasia and a CD4+/CD8+ T cell ratio of 0.8. Finally, PET/CT three months after CAR T reinfusion suggested a metabolic complete remission. However, the PET/CT five months after CAR T reinfusion revealed intensive metabolic activity surrounding the sacrum, the left femur and, again, in the left breast (Figure 2). Breast biopsy showed no evidence of lymphoma at this time point but verified invasion by CD3 positive T cells of CAR T origin. Flow cytometric analysis of cerebrospinal fluid showed the presence of CD19+/CD20+ aberrant B-cells with kappa light chain restriction in 9.3% of all cells confirming CNS relapse of mantle cell lymphoma, clinically with progressive motoric weakness in both legs. The presence of CAR T cells was confirmed by flow cytometric analysis of CNS fluid, which revealed cells with CAR T phenotype in 4.4% of all cells (Appendix A). MRI scans of the brain and spinal cord revealed focal enhancement of the cauda equina and left S1 root (Figure 3).

A prompt treatment with the bispecific CD20/CD3 antibody glofitamab was initiated together with a single dose of preceding obinutuzumab to mitigate the risk of tumor lysis and CRS and ICANS, as prescribed by the manufacturer. Despite a marked increase in serum IL-6, (Figure 1B) glofitamab treatment was tolerated without complications, and, in particular, no CRS or neurotoxicity were observed.

The expansion and persistence of CAR T cells in the peripheral blood over time since Tecartus^®^ CAR T infusion was monitored by specific ddPCR [15] (Figure 1A). Before glofitamab infusion, CAR T DNA was detected at a steady level of around 150 copies per µg DNA. Remarkably, after glofitamab administration, we observed a more than 20-fold increase (from 193 copies/µg DNA to 3909 copies/µg DNA). Importantly, the patient showed continuous improvement of motoric weakness and regained self-independency, and an MRI scan two months after the first glofitamab infusion confirmed an objective response (Figure 3). The patient completed the planned 12 glofitamab administrations without delays of the interval or need for dose reductions. At the last assessment after the planned end of glofitamab treatment, clinical remission was ongoing. After the initial peak, CAR T copy numbers quantified from peripheral blood slowly decreased to pre-glofitamab levels. The flow cytometric analysis of lymphocyte subsets is summarized in Appendix A. While the patient remained lymphopenic (absolute lymphocyte count 0.11–0.25 × 10^9^/L), the percentage of T cells increased slightly from between 21 and 28% pre-glofitamab to 40% after glofitamab administration and returned to prior levels within 4–6 weeks after the first glofitamab dose. Interestingly, we observed an increase of the CD4/CD8 T cell ratio from 2.71 before the first dose of glofitamab to 5.0 at 6 weeks after glofitamab. The CD4/CD8 ratio returned to lower levels thereafter (Figure 1C). Serum IL-6 levels showed a marked increase after the first dose of glofitamab and returned to previous levels soon after (Figure 1B).

### 4.2. Patient #2

A second female patient was diagnosed with stage IVB blastoid variant of mantle cell lymphoma with extensive involvement of the bone marrow at age 73. Flow cytometry of the bone marrow at initial diagnosis revealed a CD19+, CD20+, lambda+, CD5+, CD10-, CD79b+, CD22-, CD23- immunophenotype. She achieved a partial remission after six alternating cycles of R-CHOP and R-DHAP treatment. The patient declined consolidation with HDCT/ASCT and was treated with rituximab maintenance for eight months, when a PET/CT scan suggested progressive retroperitoneal lymphadenopathy and splenomegaly. The patient had a partial remission of lymphadenopathy to subsequent ibrutinib treatment, but had a second relapse with progressive lymphadenopathy and lymphocytosis after eight months. Flow cytometry of the peripheral blood confirmed an identical immunophenotype of the lymphoma cells as documented at initial diagnosis with expression of CD19+, CD20+, lambda+, CD5+, CD10-, CD79b+, CD22-, CD23-. B and T cell subsets are summarized in Appendix A.

The patient underwent CAR T treatment with Tecartus^®^ within the manufacturer’s early access program after bridging therapy with rituximab, bendamustine and cytarabine. Again, CAR T treatment was well tolerated without clinical evidence of CRS or ICANS. The patient achieved a complete remission as assessed by bone marrow biopsy two weeks after CAR T reinfusion. Bone marrow flow cytometry showed no evidence of MCL, complete B cell aplasia and a CD4+/CD8+ T cell ratio of 0.5 (see also Appendix A). A metabolic complete remission was documented in a PET/CT three months after CAR T reinfusion. Finally, PET/CT six months after CAR T reinfusion indicated an extensive nodal and extranodal (pleural and peritoneal) relapse of lymphoma (Figure 4), with the immunophenotype being identical with the previous diagnosis of MCL.

We started rescue treatment with glofitamab after a single cycle of salvage R-bendamustine chemotherapy, again, after administration of a single dose obinutuzumab. Again, we observed a near 20-fold increase of circulating CAR T cells in the peripheral blood as assessed by specific ddPCR (from 22 copies/µg DNA to 389 copies/µg DNA, Figure 1D). In this patient, the CAR T DNA levels remained elevated 15-fold compared to before glofitamab. As in the first patient, we noted a marked increase in serum IL-6 levels, however no relevant CRS or neurotoxicity occurred (Figure 1E).

Further clinical course was aggressive and the patient developed motor weakness in the left leg due to tumor infiltration of the left sided lumbar neuroforamina. Radiotherapy was initiated and led to tumor shrinkage and improvement of the paresis. Despite this, the condition of the patient deteriorated and the patient declined further treatment. She succumbed to disease progression and pneumonia around three months after glofitamab administration.

## 5. Discussion

Bispecific antibodies have shown promising efficacy in B-cell lymphomas, but data on the use of these antibodies after CAR T therapy are very limited. Whereas most patients in the phase I studies of glofitamab were heavily pretreated and most were refractory to the previous treatment line (90.6%), only a very small proportion of these patients was previously treated with CAR T cells (2.9%) [11,22]. Bannerji et al. reported a CR rate of 27% in 30 patients with DLBCL treated with odronextamab after CAR T failure [10]. Schuster et al. reported on the use of mosunetuzumab, another CD20/CD3 bispecific antibody, in patients with poor-prognosis B cell lymphomas [23]. In their analysis, the authors included 30 patients relapsing after CAR T therapy. Out of 18 patients with at least 3 months of follow up, seven (38.9%) achieved a partial, and four (22.2%) a complete, response. The authors identified a marked increase of peripheral CAR T DNA after treatment with mosunetuzumab. However, complete remissions were seen in these patients irrespective of CAR T expansion. Mosunetuzumab was also tested in a phase I/II study in patients with follicular lymphoma, which showed deep and durable remissions (ORR 79%). In this analysis, 3.3% of patients were previously treated with CAR T therapy [9]. Finally, in a recent report by Rentsch et al., an impressive response rate was observed with glofitamab treatment in patients with diffuse large B cell lymphoma who relapsed after CAR T therapy [14].

Bispecific antibodies form a link between a target antigen on tumor cells (CD20 in the case of glofitamab, mosunetuzumab and odronextamab) and CD3 on T cells. Noncovalent linking of CD3 to the T cell receptor leads to T cell activation without the need for costimulatory molecules, such as CD28, or proinflammatory cytokines, such as Interleukin (IL)-2 [24]. This triggers the release of cytotoxic granules, cytokines and T cell proliferation, ultimately resulting in tumor cell elimination [25,26,27]. Bispecific antibodies may, therefore, have the potential to reactivate T cells which may have been exhausted after long-term antigen exposure [28]. The mechanisms involved are poorly understood. One possible explanation is the formation of immunological synapses between T cells and tumor cells, where multiple T cell receptors are assembled, leading to amplification of stimulatory signals [24]. TCR signal strength has been shown to be of importance for T cell activation, proliferation, and cytokine production [29,30,31]. The antibody’s inherent high-affinity design [32] may result in stronger T cell activation than conventional activation by antigen recognition. This may, in part, explain the observed expansion of T cells, including CAR T cells.

Similar to our observation in two patients with relapsed MCL, Rentsch et al. demonstrated in relapsed DLBCL patients that administration of glofitamab can lead to re-expansion of CAR T cells. However, no correlation between peak expansion and survival was found [14]. The same observation was also described by Schuster et al. [23]. Thus, it remains unclear whether the bispecific antibodies, the increase of CAR T cell numbers or obinutuzumab are responsible for the beneficial clinical effects.

Relapse of B cell malignancies after CD19-directed CAR T therapy can be categorized into an antigen-positive or antigen-negative pattern. While antigen-positive relapse is closely correlated with CAR T cell exhaustion, antigen-negative relapse occurs in the context of deletion or mutation of the CD19 gene or downregulation of CD19 protein expression [33,34]. Our patients both showed a CD19-positive relapse indicating poor CAR T persistence as a possible explanation for disease progression after CAR T therapy. Consistent stimulation of CAR T cells by high antigen burden or antigen-independent clustering of the CAR receptor can lead to CAR T cell exhaustion, limiting expansion and lysis of target cells [35]. Transient downregulation of CAR protein expression, e.g., by the multikinase inhibitor dasatinib has been shown to restore antitumor activity of CAR T cells in murine xenograft models [36,37].

The observed increase of CAR T cells after administration of glofitamab may, in part, be a surrogate marker for enhanced T cell proliferation in general. Analysis of lymphocyte subsets by flow cytometry revealed a slight increase of T cell percentage and a marked increase of CD4/CD8 T cell ratio (Appendix A and Figure 1C). As no clonality analysis was performed, it remains to be elucidated whether the CAR T cell proliferation after administration of bispecific antibodies is of mono- or oligo-clonal nature. More data are clearly required on the use of bispecific antibodies after CAR T therapy to further clarify these open questions. Moreover, the optimal sequence of, and interval between, CAR T cell and bispecific antibody therapy remains to be explored. A French phase II study (NCT04703686) investigating glofitamab in patients with R/R B-Cell lymphomas after CAR T therapy is currently recruiting.

While patient #1 regained self-independence after glofitamab therapy and an objective response was observed, the second patient showed rapid progression of lymphoma and ultimately succumbed to her disease after declining further treatment, despite also showing an expansion of CAR T cells. Recently, it has been shown that higher strength of TCR signaling can result in T cell dysfunction and escape of tumor cells [38]. This may explain disease progression, despite enhanced CAR T cell numbers, in the second patient.

Obvious limitations to this report are the small number of patients allowing only for a descriptive analysis, while many questions remain unanswered. For example, CAR T expansion was only observed after the first infusion of glofitamab, but not the subsequent applications. This may, in part, be explained by the time points of CAR T ddPCR assessments. As the first glofitamab infusion was done as an inpatient procedure, assessments were done more frequently than after the following administrations. However, it also raises the question of the role of obinutuzumab in the expansion of CAR T cells and the observed clinical responses. Obinutuzumab, a second generation CD20 antibody, is administered prior to glofitamab to mitigate the risk of tumor lysis and CRS [39]. Glofitamab and obinutuzumab both bind to the same CD20 epitope [11], therefore, allowing obinutuzumab to attenuate the possible adverse effects of glofitamab during the first administration.

Both patients were heavily pretreated with rituximab and had their last administration of rituximab within 6 to 8 months prior to glofitamab therapy. Rituximab leads to B cell depletion within 72 h after administration that lasts between 6 and 9 months [40]. As flow cytometry data demonstrated (Appendix A), both patients were in B cell aplasia at the time of glofitamab initiation, pointing to a lasting effect of rituximab. Despite binding to an overlapping epitope on CD20 [41], obinutuzumab has been demonstrated to elicit stronger immune responses than rituximab, possibly due to the higher affinity of its Fc region to the FcγRIII receptor on immune cells [42], which could be an explanation for the observed high percentages of NK cells (Appendix A). However, a meaningful comparison of the clinical effects is hindered by the higher dose at which obinutuzumab is administered (1000 mg), when compared to rituximab (375 mg/m^2^). Garcia-Munoz et al. observed a significant *decrease* in CD4+ and CD8+ T cells after administration of single-agent obinutuzumab in patients with CLL [43]. We, therefore, believe that while obinutuzumab may have contributed to the clinical responses, it does not explain the marked expansion of CAR T cells seen after glofitamab administration.

CAR T therapy has revolutionized the treatment of patients with R/R B cell malignancies offering unprecedented and durable response rates. Along with the growing number of therapeutic options and CAR T products being approved for additional indications, patients with refractory disease after CAR T therapy are representing a rapidly growing clinical challenge. While bispecific T cell engaging antibodies, such as mosunetuzumab or glofitamab, have shown promising activity in R/R B-cell lymphoma, data on their use after CAR T treatment remain limited so far. Similar to the two previous reports by Rentsch et al. and Schuster et al., our patient cases suggest that administration of CD20/CD3 bispecific antibodies can support a subsequent expansion of CAR T cells. However, due to the small patient numbers, it remains to be clarified whether CAR T cells, the bispecific antibody, the combination and/or additional factors contributed to the clinical responses we observed in our patients, and any correlation between CAR T cell levels and clinical response remains speculative. Further studies may also provide novel insights into the optimal sequence of these therapeutic options.

## Figures and Tables

**Figure 1 cells-11-02747-f001:**
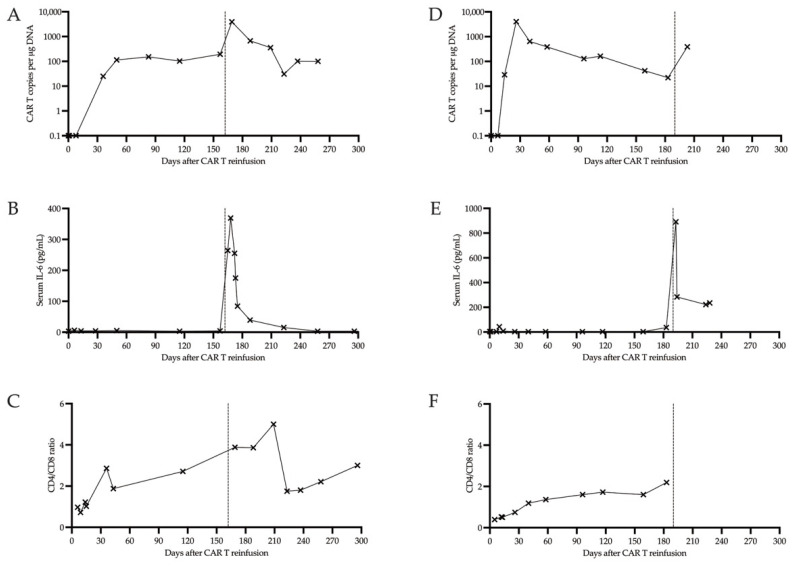
Course of parameters assessed before and after glofitamab initiation, which is indicated by a dashed line. **Panels** (**A**–**C**) **show the course of patient #1**. We observed a marked increase in CAR T DNA (panel **A**), Serum IL-6 (panel **B**) and CD4/CD8 ratio (panel **C**) after the first administration of glofitamab. **Panels** (**D**–**F**) **show the course of patient #2**. Again, we observed an increase in CAR T DNA (panel **D**) and elevated serum IL-6 levels (panel **E**) after the first administration of glofitamab. Unfortunately, due to the patient declining further active treatment, no further flow cytometric assessments were performed (panel **F**).

**Figure 2 cells-11-02747-f002:**
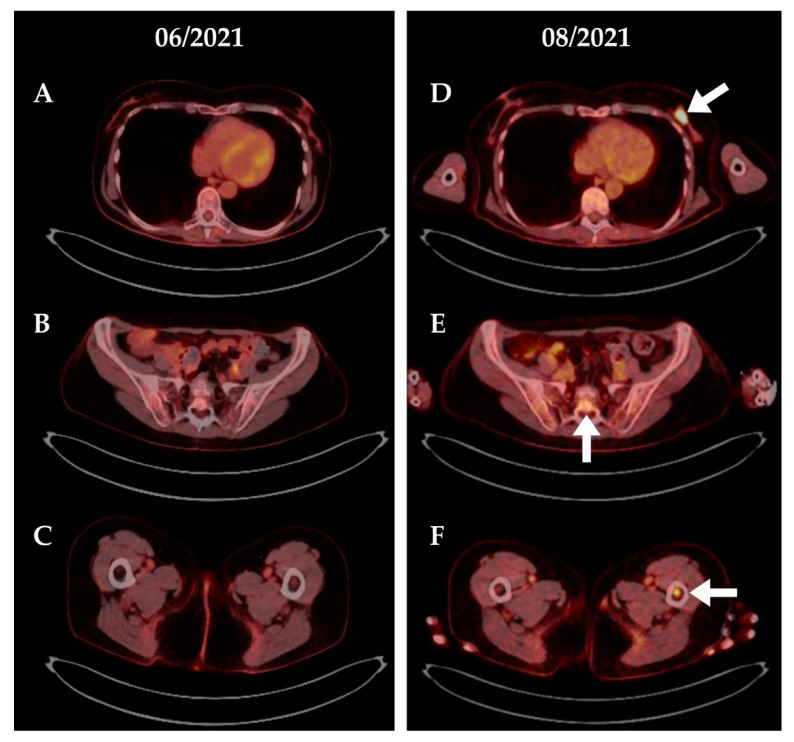
**PET/CT scans of patient #1. Relapse after CAR T therapy.** Three months after CAR T reinfusion, the patient had a complete remission without evidence of pathologic metabolic activity (panels **A**–**C**). Five months after CAR T therapy, the patient relapsed with new lymphoma manifestations (white arrows) in the left breast (panel **D**), the sacral canal (panel **E**) and the left femur (panel **F**). Biopsy of the left breast showed extensive infiltration of CD3+ T lymphocytes but no conclusive evidence of mantle cell lymphoma. Flow cytometric analysis of cerebrospinal fluid revealed evidence of CD19+, CD20+, CD5- clonal B cells confirming CNS relapse of mantle cell lymphoma.

**Figure 3 cells-11-02747-f003:**
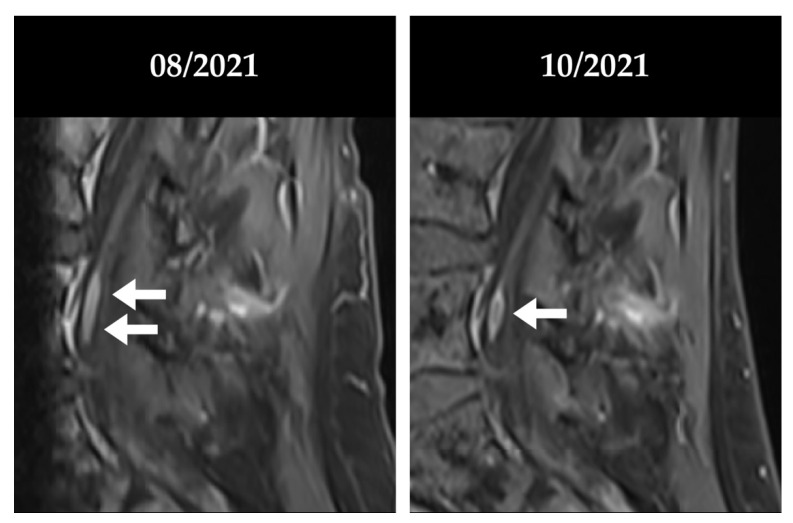
**MRI scans of patient #1. Response to glofitamab**. (**Left**): T1 dixon sequence obtained 5 months after CAR T therapy demonstrating enhancement left S1 root (two white arrows). (**Right**): T1 dixon sequence obtained around two months later, after initiation of glofitamab treatment revealing receding enhancement of the S1 root (one white arrow).

**Figure 4 cells-11-02747-f004:**
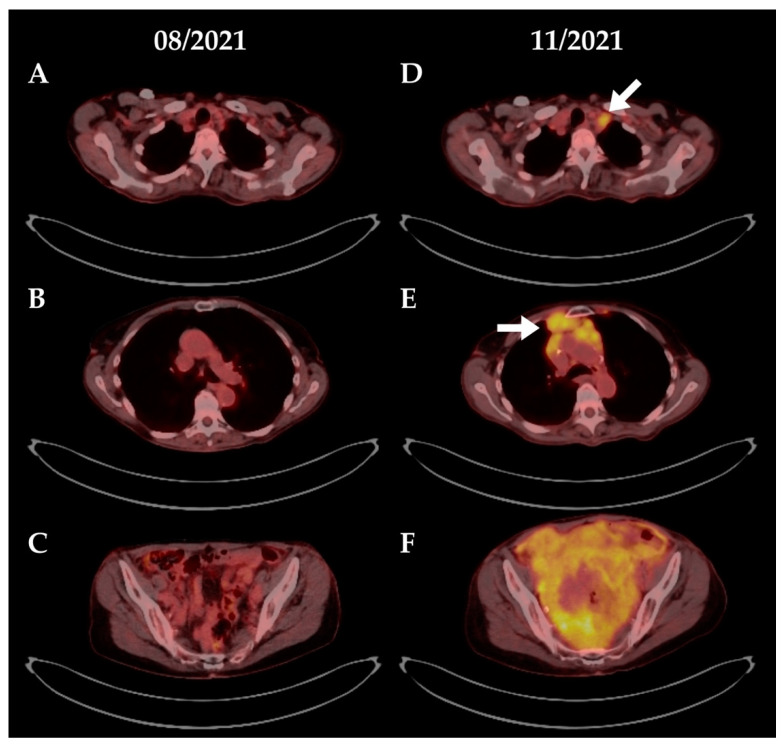
**PET/CT scans of patient #2. Relapse after CAR T therapy.** Three months after CAR T reinfusion, the patient achieved a complete metabolic remission (panels **A**–**C**). Six months after infusion, PET/CT revealed extensive nodal (panel **D** and **E**, white arrows) and diffuse extranodal (pleural and peritoneal, panel **F**) lymphoma manifestations. Treatment with R-bendamustine and then glofitamab was initiated. However, the patient denied further treatment and succumbed to pneumonia and disease progression around 3 months after glofitamab administration.

**Table 1 cells-11-02747-t001:** Characteristics and time course of previous treatments of the two patients with MCL treated with glofitamab after relapse after CAR T therapy.

Parameter	Patient #1	Patient #2
**Sex**	Female	Female
**Age at diagnosis**	56 years	73 years
**BM involvement at diagnosis**	No	Yes, 90%
**Disease stage at diagnosis**	IIB	IVB
**Therapy prior to glofitamab**	**02/17 Diagnosis** 03/17-06/17 R-CHOP/R-DHAP 07/17 HD BeEAM/ASCT 11/17-05/19 R maintenance **05/19 1st relapse** 06/19-10/19 bortezomib/ibrutinib 10/19-12/20 ibrutinib maintenance **12/20 2nd relapse** 01/21-03/21 R-bendamustine 03/21 FluCy/CAR T **08/21 3rd relapse** 09/21-05/22 glofitamab	**06/19 Diagnosis** 07/19-12/19 R-CHOP/R-DHAP 01/20-06/20 R maintenance **07/20 1st relapse** 08/20-01/21 ibrutinib **02/21 2nd relapse** 02/21-04/21 R-BAC 05/21 FluCy/CAR T **11/21 3rd relapse** 12/21 R-bendamustine 12/21-02/22 glofitamab
**Bridging therapy before CAR T**	Yes	Yes

Abbreviations: R-CHOP/R-DHAP is considered standard first line therapy in MCL and includes a total of six alternating cycles of rituximab, cyclophosphamide, doxorubicin, vincristine and prednisone and rituximab, dexamethasone cytarabine and cisplatin chemotherapy (three each); HD BeEAM/ASCT: High dose chemotherapy with BCNU, etoposide, cytarabine and melphalan followed by autologous stem cell transplantation; R: rituximab; R-BAC: rituximab, bendamustine and cytarabine; FluCy/CAR T: Fludarabine and cyclophosphamide conditioning before CAR T therapy with brexucabtagene autoleucel.

**Table 2 cells-11-02747-t002:** Summary of patient characteristics and disease course after CAR T therapy and response to glofitamab therapy. * The second patient declined further therapy with glofitamab before response assessment and succumbed to disease progression.

	Patient #1	Patient #2
**Time from CAR T therapy to relapse**	5 months	6 months
**Age at first glofitamab administration**	60 years	75 years
**Time to maximum CAR T expansion after glofitamab administration**	6 days	7 days
**Maximum CAR T expansion after glofitamab administration**	20-fold	19-fold
**Best response after glofitamab**	PR	n.a. (PD) *

Abbreviations: PD: progressive disease; PR: partial remission.

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
