# Peer review of "Experiences with Glofitamab Administration following CAR T Therapy in Patients with Relapsed Mantle Cell Lymphoma"

_cells, 2022, doi:10.3390/cells11172747_

Round 1

Reviewer 1 Report (Previous Reviewer 2)

1. English: “remission following first-line treatment are” either “remission… is” or “remissions …are”.

2. Abstract and the beginning of Introduction are virtually identical. Please consider re-writing one or another, so that the readers don’t have to read the same text twice. In my opinion, lines 13-22 provide a background information, which is excessively generous (space-wise), whereas the factoid – the results of the study - is limited to lines 23-29. Do we even need to know about 93% response rate in the pivotal trial, while reading the abstract, for instance?

3. Lines 41: “has shown to offer” doesn’t sound right to me.

4. It appears incorrect or inaccurate to call Glofitamab a BiTE. BiTE is a class of bispecific antibodies consisting of two single-chain variable fragments (scFvs) and is a registered trademark of Micromet AG (fully owned subsidiary of Amgen Inc) (see  https://jhoonline.biomedcentral.com/articles/10.1186/s13045-021-01084-4). The structure, size and PK of Glofitamab are very different from those of a conventional BiTE, such as blinatomumab. Please consider revising and correcting the terminology throughout the text.

5. Line 48: “CD3, a protein complex” sounds somewhat awkward to me. CD3 proteins are indeed part of the TCR complex. Please check if all BiTEs created to date recognize a specific subunit CD3e or an epitope formed by CD3e/d rather than simply “CD3”.

6. Line 60: move up the disambiguation of DLBCL from Line 72.

7. Line 89: “effector CD28 and the costimulatory CD3ζ”. It’s the other way around. CD28 functions as a costimulatory sequence. CD3z is an activation domain.

8. Line 97 “was containing” --> “contained”?

9. Line 104. “flexible Anti-Biotin fluochromes” --> “anti-biotin-PE conjugate”? PE-labeled anti-biotin antibody?

10. Lines 105-106. The tube did not contain CD45, CD3, CD4 or CD8. Rather, it contained the antibodies to these surface markers. Most likely, fluorescent conjugates of the antibodies.

11. Lines 282-283. I wouldn’t call IL-2 a co-stimulatory molecule.

12. Line 284. “resulting tumor”--> “resulting in tumor” ?

13. Line 309. “has shown to restore” -->“was shown to restore”? “has been shown to restore”?

14. Line 310. No need to capitalize Xenograft. Also, please consider adding an earlier reference https://pubmed.ncbi.nlm.nih.gov/31270272/ to illustrate the use of dasatinib.

15. Line 312 “which is also supported by the observed increase in T cell percentage”. I do not see that this statement is robustly/explicitly supported by the supplementary table data. To my eye, T cell percentage stays more or less the same post-glofitamab as pre-glofitamab. The argument that the increase from 21-28% to 40% is significant appears a little speculative. I am furthermore puzzled by the numbers presented in the supplementary tables. The percentages of T, B, and NK cells are provided as a fraction of all lymphocytes. Summing up the numbers virtually never results in 100% (see, for instance 58%+28%+0% on day -47 pre-glofitamab), which makes me wonder whether the apparent dynamics of T cell percentage is merely a question of math (a different denominator or, alternatively, a decreased percentage of some population of lymphocytes, which creates a false impression of increased T cell %), or an error. Please clarify. Although hematology is not my area of research, I am also wondering whether the ~50-60% NK cell levels observed are OK, and why the authors do not comment on this. Formally, one could say that such numbers could also contribute to the response, even though they were observed long before glofitamab.

16. Please harmonize the use of Glofitamab vs glofitamab throughout the text. I would stick to “glofitamab”.

17. At present, the major problem I have with the manuscript (and the title in particular) is that it pushes the glofitamab as the sole acting agent of the tumor response and CAR T cell expansion, whereas the data supporting this are missing, as admitted by the authors in their response. Even though I am inclined to agree that it is glofitamab that does most of the job, the lack of data makes this statement misleading, and it should be toned down. In terms of the tumor response, it could well be glofitamab, it could be obinutuzumab, it could be re-expanded CAR Ts, or it could be the combination of any of these factors. Similarly, CAR T cell expansion could’ve been caused by glofitamab directly, or, for instance, via an altered tumor microenvironment which has become more permissive for CAR T cells. In this regard, possible contribution of obinutuzumab would be questionable, however it couldn’t be excluded. Also, it would be expected (although not granted), that CAR T cell population expansion would follow the pattern of glofitamab infusion, i.e. would peak each time glofitamab is given to the patient. However, what we see is the opposite: a single CAR T cell peak, which happened following glofitomab AND obinutuzumab administration, suggesting the latter may have had a decisive role. It is my understanding that the timepoints for CAR T cell measurements weren’t particularly correlated with glofitamab infusion schedule or dense enough to see the additional CAR T cell peaks even if those were actually present. I have to take this as yet another weakness of the study, besides the authors’ failure to save the blood samples for later FACS reassessment(s).

All I’m saying is that all this has to be adequately and accurately discussed in the manuscript. Also, I think it would be interesting to explore whether the epitopes of rituxi, obinutuzu, glofita, and of the Miltenyi anti-CD20 antibodies used for CD20 detection on the tumor cells overlap or not. If they don’t, this information may provide additional clues why one agent, but not the other, may have had a therapeutic activity, and could also be discussed. As far as I’m concerned, glofitamab anti-CD20-binding regions are the same as those of the obinutuzumab. Both molecules have an Fc-part, although glofitamab’s Fc region is known to be completely FcRg- and C1q-dead, whereas obinutuzumab’s Fc is functional, i.e. it could well be imagined to activate FcR-expressing cells such as NK cells (see my comment above regarding high NK cell percentage) and drive/initiate tumor response.

I suggest to add a compact section “Limitations of the study” and to transparently and thoroughly discuss the above-described issues, particularly given the routine practice of giving obinutuzumab before glofitamab, which was naturally beyond the control of the authors and the scope of the study.

18. Tables 1S and 2S: “Lymphcyte”-->”Lymphocyte”

19. Again, I am not a hematologist, but to me the use of the “G/L” as a measure of cell counts is somewhat non-standard, and I would suggest using a 10[9] cells/L or similar intuitively acceptable units.

Author Response

Reviewer 2 Report (Previous Reviewer 3)

In this resubmission, authors have adressed many of the points raised in earlier review rounds. Please find my minor changes below. 

General

-        1.  Please re-allign the figures, I can’t make out half of them.

-        2.  Please re-number the figures so that they are located in order in the text, starting from figure 1 (not figure 4 as is now the case).

-        3.  Please make figure s1 look a bit more polished – now it’s a screenshot of the FACSuite software with names of populations in German.

   Specific

Line 97 – lymphoma panel was used containing

Line 100 – please mention the versions of the software used

Line 101 – as previously described

Line 109 – FACSuite software was used as previously described

Line 211 – add a reference to table s2 at the end of this sentence.

Author Response

This manuscript is a resubmission of an earlier submission. The following is a list of the peer review reports and author responses from that submission.

Round 1

Reviewer 1 Report

Mechanisms of lymphoma relapse after chimeric antigen receptor T-cell (CAR T) therapy and strategies for further therapeutic support of patients are undoubtedly among the priority tasks of modern oncology. The manuscript describes interesting clinical cases, further study of which may be extremely valuable for unlocking the potential of bispecific antibodies in the fight of CAR T treated relapsed lymphomas. The manuscript is well written and structured. Despite this, the evidence base of the work is presented by one experimental method and may lead to false interpretation of results. Additional data, including blood cell immunophenotyping by flow cytometry, serum cytokine profiling or detection of the sequences of retroviral vectors, could help to determine the likelihood of CAR T proliferation after 6 months of CAR T reinfusion.

Reviewer 2 Report

Overall, this is a nice manuscript describing a potentially transformative treatment modality for heavily pre-treated MCL patients who have failed previous CAR therapy. I believe this is an interesting work, even though glofitamab was used in MCL patients failing BTK treatment (and should be cited (https://ashpublications.org/blood/article/138/Supplement%201/130/478047), although not in the context of CAR failure, and despite the earlier publication by Schuster et al.  describing the use of mosunetuzumab in this patient category. In its present shape, the manuscript suffers from the lack of mechanistic data, being mostly descriptive.

With this in mind, there are several minor things that may need adjustments.

1) I would like the authors to include the immunophenotyping data for MCL samples throughout the course of disease (CD19, CD20, CD22, CD5, etc) to see whether the response to glofitamab was in any way predictable or, on the contrary, surprising.

2) If the samples are available, please include the FACS data  for CAR T and endogenous T cells before and after glofitamab (exhaustion markers, subpopulation make-up (Teff, Tscm, Tn, etc))

3) ddPCR data are OK to support the CAR T expansion statement, however I believe complementing these data with FACS plots would further strengthen the case

4) please comment on the possible contribution of obinutuzumab for the success of the treatment of the first patient

5) Including the data for B cell dynamics/B cell aplasia during the therapy would be of interest

6)  Is there any evidence of an oligo/monoclonal nature of CAR T cell response following glofitamab administration?

7) line 145 "Moreover, BiTE antibodies have the potential to reactivate T cells, which were exhausted after long-term antigen exposition (15)":

a) exposition--> exposure?

b) ref 15 does not specifically describe the mechanism of exhausted CART or T cell reactivation by BiTEs. Please elaborate, or, better, show CAR T cell immunophenotyping data (point 2 above)

8) One of the key biological characteristics defining the prognosis of MCL is the presence of TP53 mutation or p17 deletion. If these data are available, please incorporate them to the case descriptions.

Reviewer 3 Report

In this brief case report, Heini et al report on two mantle cell lymphoma cases treated with CAR T cells and a CD20/CD3 engager. As the current litature is limited in evidence of this type of treatment, this brief case report can be interesting for clinicians facing MCL patients relapsing after CAR T treatment. I would urge authors to add some of the patient-derived data now described in the text (i.e. scans and flow cytometry data) to the manuscript for a complete image. However, currently, there is already some data in the manuscript, but no materials and methods section or a reference to the used method is present. This needs to be added before this manuscript can be considered for publication.

Major comments

  • Please include more references. For instance, the first two sentences of the introduction should be supported by published literature. Please look critically throughout the manuscript which statements should be supported by referring to earlier work.
  • Please add (some of) the primary data for a more complete manuscript, such as some of the discussed scans and the flow cytometry data.
  • For any data described in this manuscript (for now just the ddPCR data), the method of obtaining the data and used materials need to be added.

Minor comments

General comments

  • In the title, authors use CD20/CD3 to indicate bispecificity, in the text CD3/CD20. Please change all instances to one variant for consistency.
  • Some abbreviations are only used once upon abbreviation and then no longer (i.e. R-BAC). Please remove abbreviations that haven’t been used much.

Specific comments

  • Line 47 – please very briefly explain what BiTEs are for clarity of less expert readers
  • Line 52 – please change to “a 60-year-old female”
  • Line 72 – needed to be = was, or remove this part of the sentence completely as it doesn’t add information.
  • Line 78 – please change to “The presence of CAR T cells was also confirmed by…”Line 164 -
  • Line 93 – need of = need for
  • Figure 1 – please change mcg DNA to mg (with mu sign) DNA
  • Line 136 – who had = with
  • Line 141 – where it = which
  • Line 143 – please change to “… which leads to T cell activation without…”
  • Line 146 – remove redundant comma
  • Line 146 – change exposition (?) to exposure
  • Line 148 – suggest to change to “However, it remains unclear whether the BiTE antibodies, the increase in CAR T numbers, or both are responsible for…”
  • Line 154 – please change furthermore to something such as “Therefore”, as it is rather a concluding statement, and not a continuation of a summary with new facts/points. Also suggest to change “we need more data” to “more data is required”
  • Line 156 – suggest to add a comma 2 times, after “sequence of” and “interval between”
  • Line 162 – suggest to change to “…patients with refractory disease..”
  • Line 164 – R/R has not been defined.
  • Line 165 – please change “as…..” to “Similar to a previous report by…” as it’s more correct.
  • Line 176 – this ethics statement is not sufficient. This brief report contains data from human subjects, and thus should contain a statement as to consent and/or ethical board agreement etc.

Round 2

Reviewer 1 Report

Presented results are insufficient to confirm the conclusions, the answers to the questions are not complete.

Reviewer 3 Report

I would like to thank the authors for the improvement of the manuscript. I do think this manuscript would benefit from a full length version, incorporating the supplementary information into the main manuscript, and adding the flow cytometry data with the immunophenotyping, as also requested by other reviewers and now incorporated into the text only. I don’t really think that that is fitting – this is the type of data that needs to be seen, and not just described in the text. I would like to urge the authors to mature this manuscript to a full manuscript by adding the flow cytometry data; while it would still serve as a (two) case report, it could help other clinicians make informed decisions when all data is available to peruse. Please also find below my specific recommendations based on the latest version.

 Line 47 – Remove the space between CD and 19

Line 48 – CD3 is not an antigen, it’s a protein complex that is instrumental for TCR-mediated T cell activation, which is also the reason why it’s being used for BiTEs. Please rephrase to something more fitting. This would also link it much better to the new (excellent) added paragraphs in the discussion.

Line 49 – please add that mosunetuzumab, odronextamab and glofitamab are all CD20-targeting BiTEs for clarity.

Line 55 – Please hyphenate follow-up.

Line 183 – TCR signal strength is also of prime importance for the activation of T cells, and has effects on e.g. T cell cytokine production (doi: 10.4049/jimmunol.1801026, 10.4049/jimmunol.1901167) and T cell proliferation (doi: 10.1073/pnas.1413726111) – the BiTEs have been designed with a high affinity in mind, thus probably resulting in stronger T cell activation compared to conventional TCR triggering through antigen recognition, perhaps this could also contribute to the temporal effect observed after glogitamab treatment. Indeed, recently, it was shown that high TCR signal strength results in enhanced T cell dysfunction and tumor escape (doi: 10.1084/jem.20201966). I think the manuscript could benefit from adding a paragraph or two about this.

Line 187 – what about reversal of exhaustion?

Supplementary information

Table S1 – please put table name and description above the supplementary methods and describe all used abbreviations.